# Understanding the Environmental Security Perceptions of the European Union's Security Actors

**Dimitrios Kantemnidis \* and Iosif Botetzagias** 

Department of the Environment, University of the Aegean, 81100 Mytilini, Greece; iosif@aegean.gr
\* Correspondence: kantemnidis@env.aegean.gr

**Abstract:** In this paper, we are interested in assessing the different environmental security concepts, policies, and actions of actors involved in the Common Security and Defence Policy (CSDP). Through exploratory qualitative research, we interviewed key stakeholders who formulate both the climate security discourse and its policy outcomes. Aiming at finding the different perceptions and practices among various actors, we conducted 27 qualitative interviews with practitioners from 17 different institutions, such as EU bodies and agencies, think tanks, and ministries of defense (MoDs). This article discusses the divergence between announced environmental and climate security strategies and policies related to their actual results. Notably, the findings indicate that the effectiveness of the political directives compared with the practices and the developed capabilities around the environment–security nexus are dependent on individual initiatives and efforts that a developing community of practitioners is attempting to carry out. Our study indicates that tailored environmental security policies and actions are needed to motivate both practitioners and policymakers to develop downstream methods and programs that are suited to resolving security-related challenges associated with environmental issues and especially climate change.

**Keywords:** environmental security; climate security; European Union; European security; CSDP; climate change

---



## 1. Introduction

Environmental security started to significantly evolve in the second half of the 1980s, notwithstanding the assertions of many academics that there are deeper roots in the relationship between environmental challenges and security. One of the oldest writings on environmental security, according to many scholars, is Thomas Malthus's "Essay on the Principle of Population", which connected population growth with environmental limitations to foreshadow the potential of conflict [1]. Several analogies made in the 1960s and 1970s connected environmental issues to scenarios that would pose a threat to the security of individuals, communities, or even entire states [2–5]. However, Richard Ullman's 1983 article "Redefining Security", which connected population growth, environmental quality, world hunger, and human rights to American national security, is regarded as the magnum opus on environmental security affairs [6].

In the European Union, security assessments began to take environmental factors into account in the late 2000s. The first EU report relating global security and climate change was published by the European Commission and High Representative Javier Solana in 2008 with the title "Climate Change and International Security" [7]. Because the EU's economic crisis had substantially overshadowed the environmental security issues, the EU actively returned to the topic with its Global Strategy and the Green Deal [8,9]. Both texts regard the EU as a key player in combating climate change, and they frequently make connections between it and the EU's security. However, the European Parliament's study "Preparing the CSDP for the new security environment created by climate change", released in 2021,

provided Europe with its first comprehensive review of the security concerns posed by global warming [10].

Despite efforts to include environmental issues in the European Union's security agenda, many analysts assert that member states and officials in Brussels still find it difficult to address environmental security issues [11]. Richard Youngs argues in his book *Climate Change and European Security* that despite some advancements in the EU's foreign policies, environmental issues, particularly climate change, are still not a major cause for concern [12]. Some experts attribute this lack of progress to three causes: a lack of resources, institutional constraints, and conceptual confusion. They also point out that, in addition to conceptual confusion, the EU's actions about environmental security certainly raise difficult normative obstacles [13]. Another disconnect between the climate security discourse and its policy outcomes was revealed in 2019 through interviews with practitioners from EU agencies [14]. The findings indicate that, even though a growing group of practitioners in the EU are integrating environmental threats into the EU's security agenda, these professionals frequently overlap and are in competition with one another. The existing literature thus suggests that despite being a pioneer in environmental politics since the end of the 1970s—especially those relating to climate change—the European Union shows a notable reluctance to integrate the environmental component into the routine tasks of its Common Security and Defence Policy (CSDP) actors [15]. The increasing rhetoric on climate security issues is far distant from concrete methods and procedures that will be able to specifically handle the climate challenges and environmental security issues more broadly.

In this study, we try to understand how the main actors active in shaping the European security agenda understand the concept of "environmental security" and their relation in promoting it through their organizations' mission and activities. To this end, we conducted 27 qualitative interviews with practitioners from 17 different institutions, including EU bodies and agencies, think tanks, and member-state ministries of defense (MoDs). Our approach differs from most of the research on environmental security, which has aimed at discovering a causal link between environmental factors and state or human security, frequently overemphasizing deterministic mechanisms [16]. It also differs from approaches that strive for reaching a universal definition of environmental security, one that can fit into the aims and objectives of different security actors and agencies, and thus overcome the existing ideational fragmentation, which they consider to be highly disadvantageous for achieving security for individuals [17].

Some scholars strive for a universal definition from an authoritative source to overcome the existing ideational fragmentation, which they believe is highly disadvantageous to achieving security for individuals [17]. On the contrary, we are interested in studying what motivates the different security actors to include the environment in their agenda in the particular ways they had done so, and what this diversity of approaches in the real world may tell us about the relevance of existing environmental security theories. Thus, undertaking this research has three aims: first, to contribute to filling the research gap on how EU policy actors perceive environmental security; second, to understand which ideational elements and how the ideational heterogeneity affect real-life environmental security policymaking; and third, to assess whether the existing different EU environmental security strategies now employed by security actors may lead to beneficial policies and, if not, how they may be ameliorated.

## 2. Literature Review

### 2.1. Historical Background of the "Environmental Security" Discourse

Since the early 1960s, many academics have portrayed a variety of environmental challenges as immediate threats to humanity and, subsequently, to nations. The urgency of dealing with these issues was frequently demonstrated with dramatic illustrations, emphasizing the need for governments and citizens to act to survive. The book *Silent Spring*, by Rachel Carson in 1962, is one such example [2]. Its core argument was that anthropogenic



environmental change must be approached with extreme caution because we are destroying the natural systems that support our very existence. Similarly, in 1968, the ecologist Garrett Hardin brought to public attention a warning first issued in the nineteenth century by the economist William Forster Lloyd on the "inherent" vulnerability of common resources [3]. Hardin popularized the concept of "global commons" in his 1970s article "The Tragedy of the Commons", where he argued how self-interested human behavior concerning the environment would lead to mutual collapse, highlighting as the prime driving force the ever-expanding human population, which at the end would surpass the earth's "carrying capacity" [18] (pp. 137–138).

The "tragedy of the commons" idea was soon followed by the "limits to growth" thesis [19] (p. 11). In 1972, the apolitical Club of Rome mobilized the Volkswagen Foundation to fund an MIT research team for its project: "On the Predicament of Mankind". The findings of this research provided the basis for the book *The Limits to Growth*, by Meadows et al., which sold 12 million copies in thirty-seven different languages. The book introduced the concept of anthropocentric climate change to a mass audience, while it made environmentalists, scientists, and policymakers think of environmental problems in large-scale terms and as dynamically interconnected [20] (pp. 8–152, 229, 391). In the *Limits to Growth* preface, the UN secretary-general, U Thant, wrote,

"I do not wish to seem overdramatic, but I can only conclude from the information that is available to me as Secretary-General, that the Members of the United Nations have perhaps ten years left in which to subordinate their ancient quarrels and launch a global partnership to curb the arms race, to improve the human environment, to defuse the population explosion, and to supply the required momentum to development efforts. If such a global partnership is not forged within the next decade, then I very much fear that the problems I have mentioned will have reached such staggering proportions that they will be beyond our capacity to control" [4] (p. 17).

U Thant went on to authorize the convening of a conference on human–environment interaction in response to a proposal from Sweden to the United Nations Economic and Social Council (UNESOC). On 5 June 1972, 113 states attended the Stockholm United Nations Conference on the Human Environment [18] (p. 136), [19] (pp. 3–7). During the conference's preparations, Canadian diplomat Maurice F. Strong, who also served as its secretary-general, formed a committee of experts led by Dr. René Dubos to prepare an informal report to influence public opinion and governments on the meeting's content. The result was the book *Only One Earth: The Care and Maintenance of a Small Planet*, by Barbara Ward and René Dubos [5]. Ward and Dubos's book was an effort that publicly debated the words "development" and "environment", expressing worries about environmental degradation and inadvertently securitizing environmental issues, similar to Rachel Carson's, Garret Hardin's, and the Club of Rome's contributions.

While the issue of "security" was lurking in all the earlier discussions about development and (its effect on) the "environment", it was in the 1980s that several actors squarely incorporated the environmental component into the security discourse, giving rise to the environmental security concept [21,22]. In 1983, Richard Ullman, in his article "Redefining Security", related population growth, environmental quality, global hunger, and human rights to United States national security, urging countries to demilitarize international relations since the aforementioned nonmilitary threats were expected to become prominent and exacerbate insecurity [6]. In a similar vein, 6 years later, Jessica T. Mathews, in an article also titled "Redefining Security", recommended a redefinition of what constitutes national security in the United States, highlighting climate extremes, the greenhouse effects, changes in the carbon and nitrogen atmospheric cycles, and environmental refugees [23]. By then, the concept of "environmental security" has found its way into high-profile actors' reports [24]. In 1983, the United Nations General Assembly established a commission, led by Norwegian ex–Prime Minister Gro Harlem, to propose ways of achieving sustainable development. Four years later, the Commission's report "Our Common Future" drew also a clear line connecting the environment to security: "Environmental stress is both a cause and

an effect of political tension and military conflict" [24] (pp. 239–240). The report also went a step further by highlighting the detrimental effects of a, then, still-debated environmental "problem", that of climate change: "Slowing, or adapting to, global warming is becoming an essential task to reduce the risks of conflict" [24] (p. 294).

While many international figures and organizations supported this newly created, broader vision of security and the increased awareness about human-caused environmental degradation, the lack of empirical data meant that the conventional notion of state-centered, national security had not been seriously challenged. All that changed in the 1990s, when several environmental security analysts presented empirical insights into the link between environmental change and conflict in response to criticisms about the lack of supporting evidence. The University of Toronto's Project on Environment, Population, and Security, led by Thomas Homer-Dixon, intended to deviate from the philosophical discussion and develop research on a firm empirical foundation [25] (p. 475). The Toronto Group's research offered evidence of the causal path from scarcity of cropland, forest, fish stocks, and water to violent conflict by using examples from developing countries [26–28]. Similarly, in February 1994, the journalist and travel writer Robert Kaplan directly challenged the conventional security agenda in his article "The Coming Anarchy". In Kaplan's own words, "It is time to understand the environment for what it is: the national-security issue of the early twenty-first century" [29]. Kaplan's dramatic illustration caught the interest of the vice president of the United States, Al Gore, who invited Homer-Dixon and the Toronto Group to consult him on environmental change and its security implications. The major finding of the group, that various forms of environmental constraints may trigger civil conflicts, prompted the Clinton administration to release federal funds for environmental security considerations and establish the position of deputy undersecretary of defense for environmental security [30] (pp. 75–76).

### 2.2. From Environmental Security to Climate Security: An Emerging Trend

Accordingly, the 1990s was a period when major security actors started to seriously engage with the "environmental threat". Thus NATO, with Javier Solana as its secretary-general (1995–1999), enriched its diplomatic tools with an environmental agenda. In 1997, NATO's Founding Act with Russia referred to the need for cooperation on defense-related environmental issues. Solana and US President Bill Clinton worked closely for 5 years and launched the Euro-Atlantic Partnership Council (EAPC). It brought together 16 NATO nations and 28 partner countries to cooperate on various environmental security-related issues [31].

The 1990s was also a period when climate change came to the spotlight of global attention. Already in 1988, the World Meteorological Organization (WMO) and the United Nations Environmental Program (UNEP) created the Intergovernmental Panel on Climate Change (IPCC) to provide objective scientific information on climate change and to present the state of scientific knowledge—with its "First Assessment Report" coming out in 1990 and thereafter updated every 5 years. In 1992, the UN Framework Convention on Climate Change (UNFCC), aiming "to achieve [the] stabilization of greenhouse gas concentrations in the atmosphere at a level that would prevent dangerous anthropogenic interference with the climate system", was signed by 154 states in Rio de Janeiro, Brazil.

Despite these developments, climate change did not feature prominently in discussions about environmental security in the 1990s. It is quite telling that the Toronto Group considered climate change as a long-term process that did not require consideration in their specialized studies on environmental change and violent conflict. For example, Homer-Dixon, who had a significant impact on the US environmental security policies throughout the 1990s, was writing in 1994 that "When analysts and policymakers in developed countries consider the social impacts of large-scale environmental change, they focus undue attention on climate change and stratospheric ozone depletion" [27] (p. 7). Thus, even though numerous scientific forecasts appeared in the 1990s concerning the severity of this particular threat, climate change did not gain traction as numerous critics

suspected that the security establishment was exaggerating "non-traditional" security risks [32].

This changed in the early 2000s when systematic academic research on climate security challenges gained traction [33,34]. The first major milestone was a 2002 study by the National Research Council (NRC)—a private, nonprofit organization—following a request from the US Global Change Research Program (USGCRP), which represents several federal agencies in the United States [33] (p. 2). It was soon followed by Schwartz and Randall's 2003 report for the US Pentagon [33,34]. Both articles examined the prospect of a sudden climatic shift and its implications for US national security, and they are regarded as the starting point for the evolution of the climate security concept [35]. Across the Atlantic, the mood was also changing. In 2004, Sir David King, the British PM's senior scientific advisor, stated, "In my view, climate change is the most severe problem that we are facing today, more serious even than the threat of terrorism" [32] (p. 415), while 3 years later, in 2007, the United Kingdom convened the first-ever United Nations Security Council summit on climate change [32].

The latter is one of the reasons for which several scholars consider 2007 to be a landmark year for climate security [16,32,35–42], the second being the fact that in that same year, IPCC, along with Al Gore, was awarded the Nobel Peace Prize. Both cases were unprecedented. It was particularly radical that the UNSC, instead of convening to address a war situation or an act of aggression, convened to address climate change as a threat to peace and security. Of the 55 speakers, 24 agreed that the Security Council was the appropriate forum for addressing climate change policy, 13 disagreed, and 18 did not take a position. Most speakers from the global South opposed the Security Council's engagement in climate change, while 70% of the speakers from the global North supported the Security Council's involvement [43]. Despite the disagreements, the British chair, with the assistance of small-island states, succeeded in a unanimous nonbinding UN General Assembly resolution [44].

Similarly, the awarding of the Nobel Peace Prize to the IPCC and Al Gore was another extraordinary occurrence that reinforced the notion of climate security. The Nobel Peace Prize is generally awarded to those who have done the most or the best work for fraternity between nations, for the abolition or reduction of standing armies, and for the holding and promotion of peace congresses [45]. While the IPCC's mission is to offer impartial scientific information on, and assessment of, climate change, starting with its Third Assessment Report (2001), it had begun to highlight the effects of climate change on sociopolitical, economic, and security levels [46]. Gleditsch and Nordas identified numerous linkages to what they called the "climate–conflict nexus" after reviewing the Third (2001) and Fourth (2007) Assessment Reports of the IPCC [20]. Thus, the IPCC gradually started to emphasize climate change as a global threat affecting all countries, indicating that global action is necessary to address these threats. The highlight of this effort was reflected in its 2015 Fifth Assessment Report, when the IPCC included a 36-page chapter titled "Human Security" [47].

The startling events of 2007 sparked several reactions yet also motivated numerous institutions to conduct assessments on the connections between climate change and security [48]. The most comprehensive report (2007) came from the Scientific Advisory Council on Global Environmental Change of the Federal Republic of Germany (WBGU), a group composed of nine prominent natural and social experts from Germany and Switzerland. In WBGU's view, climate policy would be a preventive security policy [49]. International Alert, an international nongovernmental organization (NGO) sponsored by the UK Department for International Development, conducted a less thorough and more narrowly focused research (2007) [50]. The CNA Corporation, a US Navy think tank, issued a report (2007) focusing on US national security, though human security issues were not neglected [51]. Finally, a report (2007) by the Center for a New American Security, which is largely composed of former high-ranking officials of the Clinton administration, emphasized the risks to all nations and the importance of international cooperation [52].

Today, the idea of climate security has grown considerably in the United States, both at the government level and among nonstate entities, such as institutes, think tanks, and universities. The way those actors conceptualize and articulate the concept of climate security is largely focused on how climate change affects either the homeland and national security of the United States or how climate change may threaten US interests overseas [53–58]. One of the institutes famous for its prominent climate security research is the Woodrow Wilson Center, which has incorporated climate change issues into its Environmental Change and Security Program (ECSP). The ECSP program, which was founded in 1994 to convey academic research on environmental security to policymaking audiences, is one of the longest in the field. Several Washington-based institutes are also dedicated to climate security, including the CNA Corporation, the Center for Climate and Security (CCS), the American Security Project (AMS), and the International Military Council on Climate and Security (IMCCS) [59,60].

### 2.3. The Issue of Climate Change in the Security Agenda of Europe

The European Union declared the need for a unified security and defense policy through the Treaty on the European Union on 7 February 1992, in Maastricht, a few months after the fall of the Berlin Wall [61]. This policy should address all issues relating to the Union's security, including the potential establishment of a common defense strategy that could eventually result in a collective security structure. The Common Foreign and Security Policy (CFSP) was established by the treaty as the second pillar of the new EU'S three-pillar framework. The European Security and Defence Identity (ESDI) concept was how the broader CFSP concept was put into practice. Its goals were to defend Europe in areas where the US or NATO had no interest and to give the EU more authority over its security course.

However, no substantial steps were taken towards a common defense until French President Jacques Chirac and British PM Tony Blair signed the Saint-Malo Joint Declaration on 4 December 1998 [62] due to their common understanding that it was crucial to implement the CFSP principles fully and quickly to forward the formulation of a common defense policy. This movement's central component was a quick response to emerging threats. Thus, in 1999, ESDI was renamed to European Security and Defense Policy (ESDP), which was to last for 10 years until the Treaty of Lisbon in 2009 and the creation of the Common Security and Defence Policy (CSDP) [63] (pp. 8–10).

CSDP is the first security attempt to address the CFSP's mandate and to include all questions related to the security of the European Union. The Treaty of Lisbon except for CSDP introduced some new actors to address the CFSP's objectives: the high representative/vice president (HR-VP), the council president, the European External Action Service (EEAS), and the European Defence Agency (EDA) [63] (p. 26). All these actors were supposed to set the security agenda regarding proactive and reactive responses to potential security challenges. They should also address risks and construct their missions following the European Security Strategy (ESS), which was officially revealed in 2003 and amended in 2008 [64,65].

As a pioneer in environmental politics, the European Union has made significant efforts to address climate change concerns through initiatives that are frequently connected to its security and foreign policy [7–10,66–72]. Climate security was smoothly introduced into the political agenda of the EU. The 2008 revised edition of the ESS by the European Council followed the recommendations from the high representative and the European Commission to produce the first EU paper on Climate Change and International Security [7]. Although the EU's economic crisis overshadowed the climate security issues, the EU actively returned with its EU Global Strategy and the EU Green Deal. However, for the first time since *Solana's Paper*, climate security is addressed in 2021 when the European Parliament delivered its first in-depth analysis of the security threats posed by climate change to Europe with its report *Preparing the CSDP for the new security environment created by climate change* [10].

Numerous nonstate actors also raise awareness of and conduct research on climate security threats, either independently or with governmental support. The Institute for Environmental Security (IES) was one of the earliest nongovernmental organizations to deal with climate security, and it was mainly responsible for introducing the environmental security concept into a Brussels policy. Established in 2002 by high-ranking officials of EU agencies and governments with significant influence in the European Commission and the European Parliament, the IES initiated two major programs, the 2007 Climate Change and International Security and the 2009 Climate Change and the Military, which later evolved into the current Global Military Advisory Council on Climate Change. The IES no longer has a vibrant presence in Brussels, but its place has been taken by two other institutions with significant influence in European institutions: the Stockholm International Peace Research Institute (SIPRI) and the Berlin-based adelphi [73]. All those entities have enabled the European Union to develop a climate security practice community that routinely advises European governments and agencies. This community of practice is closely related to initiatives such as the German government's Group of Friends on Climate and Security, the Dutch government's Planetary Security Initiative, the adelphi's Climate Security Expert Network (CSEN), and the Hague Roundtable on Climate and Security [60].

## 3. Methodology

An exploratory qualitative study was carried out to interpret reality and understand the effectiveness of environmental security methods and programs. The questionnaire was submitted for review and granted permission by the authors' home University Research Ethics and Integrity Committee. Furthermore, prior to each interview, the participants were given an information sheet regarding the research and signed the relevant consent form.

We employed a bottom–up approach by conducting qualitative research based on 27 semistructured in-depth interviews with policymakers from 17 different institutions, such as EU bodies and agencies, think tanks, and ministries of defense (MoDs) [74–76]. The study aimed to explore what motivates different security actors to include environmental considerations in their security policies in the manner that they did and how these heterogeneous policies may impact the EU's efforts to address climate threats. We originally intended to interview 25 persons, but we ended up interviewing 27, as some interviewees suggested other people as subject-matter experts during the process. Our selection criteria called for both civilian and military professionals at the middle and senior levels to work for organizations that set the European security agenda. Table 1 lists the positions and codes of the interviewees.

By researching how the EU's policymakers respond to the question "security from what?", "security for whom?", and "security through which means?", we can address the EU's environmental security policies [77]. The answers to these questions comprise three components that define an environmental security policy: first, the security actors' perceptions of environmental risks; second, the security actors' objectives and mandates; and third, the available mechanisms and resources to respond to environmental threats [78,79]. These three components combine to form the outcome variable, which is the response of the respective security actor, i.e., the EU's environmental security policies. For instance, climate change is seen as a severe concern by both the EDA and the ESDC (security from what?). However, the EDA's mandate (security for whom?) focuses on the security of EU citizens in connection to the defense industry, whereas the ESDC's mandate is focused on education and training. Furthermore, the EDA has significantly greater resources and means than the ESDC. That is, the ESDC policy on climate security differs significantly from the EDA policy, and it is explained by examining it considering the three points raised above.

**Table 1.** List of interviewees.

| Interview Title | Interviewee Code | Medium | Date |
|---|---|---|---|
| Environmental Security and EDA | INT-1EDA | Virtual Teleconferencing (VTC) | 2022 |
| | INT-2EDA | In Person | 2023 |
| | INT-3EDA | In Person | 2023 |
| | INT-4EDA | In Person | 2023 |
| Environmental Security and EEAS | INT-1EEAS (EUMS) | Telephone | 2022 |
| | INT-2EEAS | VTC | 2022 |
| | INT-3EEAS | In Person | 2023 |
| | Mikhail Kostarakos (EUMC) * | Email | 2023 |
| Environmental Security and ESDC | INT-1ESDC | In Person | 2022 |
| | INT-2ESDC | VTC | 2021 |
| Environmental Security and EUISS | INT-1EUISS | VTC | 2022 |
| | INT-2EUISS | VTC | 2021 |
| Environmental Security and European Commission | INT-1EC (DG ENV) | VTC | 2021 |
| | INT-2EC (DG CLIMA) | VTC | 2022 |
| | INT-3EC (DG HOME) | In Person | 2022 |
| | INT-4EC (DG HOME) | In Person | 2023 |
| | INT-5EC (DG JRC) | In Person | 2023 |
| Environmental Security and SatCen | INT-1SC | In Person | 2022 |
| | INT-2SC | Telephone | 2022 |
| Environmental Security in Europe and Ministries of Defense | INT-1MoD | In Person | 2022 |
| | INT-1MoD | In Person | 2022 |
| | INT-1MoD | In Person | 2022 |
| | INT-1MoD | In Person | 2022 |
| Environmental Security in Europe and Nonstate Actors | INT-1nSA (adelphi) | VTC | 2021 |
| | INT-2nSA (EY) | VTC | 2021 |
| | INT-3nSA (PRIO) | VTC | 2022 |
| | INT-4nSA (SIPRI) | In Person | 2023 |

\* General Mikhail Kostarakos (chairman of the European Union Military Committee, European Union External Action, 2015–2018) asked to be identified as one of our interviewees.

A semistructured interview guide with three different types of questions served as a guide for the interviews. After 8 pilot interviews, the questionnaire was created and validated. All interviews started by asking more general questions to set the stage, such as how long people had been employed in the particular organization and how they came to work for it, what duties that division or agency has, and how may that agency help with CSDP operations and missions. Additionally, we included some environmental dimensions to these questions, such as how their agency learns about environmental and climate change challenges and how much security issues arise as a result of environmental changes.

In the second part, we narrowed down and focused on three categories of questions: first, questions that reveal the potential beneficiaries of security (security referent objects), thus who should receive the political good of security; second, questions into how much policymakers consider environmental concerns to be threats, threat multipliers, or even unrelated to security issues; and third, questions about how to maintain security as they vary from one security actor to another due to various mandates and the wide variety of conceptual interpretations of the term "security". Thus, the mechanisms, resources, and interagency cooperation on environmental security challenges were the main areas of focus.

The whole process took place between November 2021 and February 2023. The interviews took place over a longer length of time than expected because it became clear during the interviews that other actors who were not initially included in the research needed to be added to acquire a complete assessment. Additionally, the constraints brought on by the COVIDepidemic frequently served as a barrier, and numerous participants found it challenging to take part in online sessions. The interviews were processed using the following methodological steps: interview coding, reception, and interpretation of find-

ings. An alphanumeric code is used to identify the participants that want to maintain their anonymity.

## 4. Results

Overall, we did not encounter any problems with the interview process. The participants shared a similar characteristic in that almost none of them had any formal academic expertise in environmental or climate change issues. When making assessments, the majority of them frequently considered personal variables and drew on the knowledge they had mostly gained from their professional or personal involvement in the subject. Additionally, it may be viewed as a minor problem that we frequently did not receive a response to our requests for interviews with the EEAS or the directorates of the European Commission.

Participants in this study were either employees of the organization with a working understanding of environmentally related topics or higher-level executives who discussed with us the organization's objectives regarding the environment. Thus, these individuals were divided into groups according to the purposes that each organization, as stated by its mandates, serves. The 27 interviewees were classified into the following five groups: European External Action Service (EEAS), European Commission, independent agencies, ministries of defense, and nonstate actors.

### 4.1. European External Action Service (EEAS)

The EEAS survey respondents emphasized the relevance of environmental issues in EU security challenges—although one of them stated that since environmental issues are low-politics issues, they are not a priority for the EU, which has a high-politics perspective. The interviewees identified the main environmental threats to Europe's security as being related to climate change effects—high temperatures, extreme weather, desertification, and rising sea levels [80]. They also underlined the necessity for a speedy recovery following natural disasters in nearby afflicted countries, such as the most recent earthquakes in Turkey [81], and considered environmental crimes, such as marine pollution, poor garbage, and waste management, as dangers to Europe's solidarity [82]. In response to the question "Are there any instances you have dealt with which provide evidence that environmental issues have a relation to EU's security?", the former chairman of the European Union Military Committee, General Michael Kostarakos, made a statement that is noteworthy; he claimed that the environmental phenomena at the Horn of Africa, where EU military personnel are carrying out the antipiracy naval Operation ATALANTA and the Training Mission in Somalia, serve as the best examples. Tens of millions of the local population are either affected by or at risk from the hunger that the climate changes have brought about. He emphasized that the pirate attacks on cargo vessels in the Red Sea and the Indian Ocean that have returned since March 2018, as well as several military raids on the mainland, should be seen as direct repercussions of weather, climate, and their potential effects [83].

All respondents adopted the same stance in response to the second set of questions about who should benefit from security concerning environmental risks. The responses suggest that, aside from European citizens, the European agencies and the member states should take proactive measures on behalf of all nations worldwide [80–82]. They specifically noted the EEAS's 140 offices and delegations around the world, which could serve as Europe's eyes, ears, and mouthpiece regarding environmental threats in the countries they are stationed in. Considering this, they contend that Europe can protect its citizens while still acting as a global security actor regarding environmental security challenges.

In response to the final set of questions about "security through which means?" and how to deal with the numerous environmental security issues, the respondents emphasized that more needs to be done. One participant, who had contributed to drafting the EU's "Climate Change and Defence Roadmap" [72], emphasized the need for the EEAS to strengthen the civilian side of environmental security issues by hiring more climate experts, lawyers with experience in environmental law, and experts in humanitarian development [81]. Another participant opined that we should establish a mechanism to oversee environmental

issues in connection with security challenges. More specifically, he asserted that he had previously made a carbon footprint methodology recommendation to his political executives that would benefit the military of the member states, the defense sector, and European decision makers [80]. Environmental education for the security community and changes to international regulations to incorporate the carbon emission from military operations in the states' overall quantity of emissions were some additional measures suggested by the interviewees of this group [82,83].

*4.2. European Commission*

A different position was taken by the European Commission policymakers' interviewees. They suggested initiatives involving more civilian actors and less of the security community as a way to address the impending environmental security issues. At the same time, in this group, we find a greater emphasis put on European deficiencies and less attention paid to the global actions that the EU should take.

In particular, with regard to questions related to the risks and challenges of the EU's security because of the environmental pressures, the respondents overall agreed with the idea that "there is an existential threat, but we need to be prepared with state and European mechanisms and less with military engagement" [84]. To the latter end, a DG ENV interviewee suggested many "defense initiatives" already financed by the EU LIFE 2021–2027 program, yet a closer look reveals none of them are related to military training, technology, or operations [85].

Climate change is the top concern for all responders since it may cause problems within European states. They claimed that CC might catalyze instability by increasing migration pressures and lowering nations' capacity to guarantee citizens' security [86]. Participants from DG HOME were particularly in agreement that the fragile political, social, and economic structures that serve each failed state will be particularly strained by climate change [84]. The public's trust in their leaders will decline if a country is unable to satisfy the demand of the population for basic needs, and this could lead to internal conflicts over resources, such as land and water, or even state collapse. Even in supposedly stable regimes, the combination of existing strains and climate change may overwhelm communities and even lead to societal unrest [87]. The DG CLIMA and DG ENV staff underlined that it is known that climate-related events enhance and amplify the likelihood of conflict, although the exact mechanisms are unclear. Thus, instead of asking whether climate change causes conflict, they were more interested in establishing how it affects all stages of the conflict cycle [85,86].

These interviewees' prime emphasis was on European concerns, and it was European citizens who were the ones expected to benefit from environmental security ("security for whom?"). Their primary focus was on the security of the European Union and on international unrest that could have an indirect impact on Europe. The sole exception came from an interviewee who had served as head of a unit active in promoting the EU's international environmental agenda in the 1980s and 1990s, who said that the EU should assume responsibility for maintaining international security and acting as a global actor. For this interviewee, the role of Europe on environmental security issues is crucial, and "the EU is the moral actor behind every environmental cooperation around the globe" [85]. On the other hand, the new generation of policymakers in the Commission was more pragmatic and more worried about the security of the European population rather than the EU's "duty" to act globally.

Except for the DG JRC interviewee, who recommended various actions for the armed forces, the measures to address environmental challenges were primarily soft-power activities. The necessity for a mechanism that could assess the impending migratory flows caused by climate change was of utmost importance to DG HOME. Thus, the EU could be ready to welcome refugees while also preparing to integrate them to not exercise pressure on local communities at the European borders [84,87]. For the interviewees of DG CLIMA and DG ENV, there exists a need to utilize the EU's Green Deal and to improve EU tools,

such as the EU Conflict Early Warning System, the EU Conflict Prevention Network, the EU Climate Defense Roadmap, and the INFORM Risk Index, through providing them with more funding and personnel [85,86]. On the other hand, the DG JRC interviewee suggested measures for the armed forces that could increase the EU's resilience to environmental security threats, such as integrating natural disasters and climate change in conflict forecasts to trace and inform of prospective security challenges [88]. Additionally, s/he proposed the need for further research on how natural disasters and climate change may affect vital military infrastructure and EU security, particularly critical energy infrastructure related to defense. The findings of this research may shape the development of a long-term strategy to promote risk mitigation and resilience building of military facilities and defense-vital infrastructure in Europe, minimizing potential losses and unforeseen events. Last but not least, s/he asserted that we must significantly lessen the environmental impact of EU defense, which has the added benefits of lowering resource dependence, raising energy performance, preventing pollution, producing cost reductions, and providing environmentally informed decisions [88].

*4.3. Independent Agencies: European Union Institute for Security Studies (EUISS), European Defence Agency (EDA), European Union Satellite Centre (SatCen), European Security and Defence College (ESDC)*

Interviewees from independent agencies responded to the set of questions in a variety of ways, and their approaches were directly related to the mandate of their organizations. Even while they all agreed that environmental challenges were crucial to the security of the EU, each subcluster proposed a different approach to dealing with them.

The EDA staff defined climate change for the European military forces as both a critical challenge and an opportunity. "EDA has always been a pioneer on green initiatives", according to a respondent who had spent 10 years with the agency before joining the private sector [89]. Another interviewee stated that "for more than a decade, the EDA has been working on projects including energy-autonomous camps for deployed battlegroups, green infrastructure for military bases, renewable energy sources for the armed forces, deployable biomass and water production systems, and research on biological effects of exposure to acoustic and electromagnetic fields" [90]. Another one stressed that the *Consultation Forum for Sustainable Energy in the Defence and Security Sector*, which has been active since 2015, involves representatives from the ministries of defense (MoDs) of the member states and plays a major role in the EDA's green initiatives. The *Consultation Forum* aspires to lower the dependency of the MoDs on fossil fuels and natural gas, gradually cut energy costs and carbon emissions, and improve operational efficiency and energy resilience [91]. The Energy and Environment Working Group is another EDA initiative that, according to the EDA's staff, is connected to the objectives of the European Commission's "Climate Change and Defense Roadmap". This group's main foci are energy security, dependence on fossil fuels, operations resource security, water security, and climate change [92]. The EDA staff claimed that they are well informed about the environmental issues that the EU must address, but they asserted that the situation is different in the member states' MoDs, where national priorities take precedence over European ones [90–92].

The personnel at the ESDC and EUISS agreed that combating climate change is a major priority for the EU as a whole and for their respective organizations. They presented their organizations as being responsible for raising awareness of the urgent security threats that the EU is currently facing, along with the pressing environmental issues [93,94]. They claimed, "Because of the global role it must play, the EU's aspirations are not limited by European borders" [90–92]. They occasionally take on several projects, some of which have environmental components that relate to both the CSDP and international security challenges [95]. In addition to the reports on environmental security challenges over the years, an EUISS interviewee indicated that, in 2021, they hired an environmental security analyst to produce ad hoc projects [96]. Similarly, the ESDC staff noted that they have a new module on climate security issues to inform the security community about those "new

and hot topics" while including scholars working on environmental security issues in their doctoral school [95].

The SatCen staff argued that while they are aware of the environmental challenges that exist concerning the EU's Common Security and Defence Policy (CSDP), they are not working at all on environmental issues. According to one of the interviewees who has worked with SatCen for many years, "While today SatCen has no environmental orientation, in the past we have worked on Haiti's earthquake and Asian tsunamis", and "we had also requested to see if a dam is damaged by ISIS and how they destroyed cultural sites" [97]. Another interviewee emphasized that although there are employees with environmental backgrounds, there are no environmental roles in SatCen, and the only involvement in "environmental" tasks occurs through the collaboration with the divisions under the deputy secretary-general for CSDP and crisis response in EEAS, which work on some environmental projects [98].

*4.4. Ministries of Defence (MoDs)*

The MoDs interviewees were approached during a conference on the EDA's "Consultation Forum for Sustainable Energy in the Defence and Security Sector" in November 2022. It was apparent that these topics were relevant to their day-to-day work and that they were familiar with the energy and environmental challenges of the security community. Some of the interviewees, however, implied that because the security community is less knowledgeable about environmental and climate change issues, their familiarity with these topics did not match their colleagues' average level of understanding [99]. By saying this, they communicated their expertise compared with their colleagues.

When asked who should benefit from European security, the armed forces personnel's views were clear. One of the main concerns being raised by every military officer was the necessity to protect the people of his country from the consequences of climate change [99–102]. Military personnel and equipment were a second object in need of protection from any environmental hazards [99,102]. The replies specifically addressed how the operational environment of war will alter in the coming years as a result of climate change and what factors defense sector manufacturers should take into account to sustain the fighting efficiency of the national armed forces [100,101].

To the second group of questions—"security from what?"—the group of officers gave a variety of responses despite the major concern being related to the side effects of climate change. One of the main issues raised by the interviewees was whether military base infrastructure will be able to endure imminent climate change and how future extreme weather phenomena will affect the theater of operations [99,101]. Because the military's current equipment is outdated and electric vehicles would increase operational effectiveness, two participants suggested that the equipment of the armed forces should be replaced and have a lower carbon footprint [100,102]. When questioned about the success chances of such a proposal, they responded that they thought "greening" military operations was simpler than doing it for civilian ones. Nevertheless, the military officers expressed concern about going public concerning the "greening" of the armed forces; for example, a suggestion to establish a platform for tracking the units' carbon dioxide emissions was frowned upon, since it could offer some useful indirect intelligence to potential enemies. The possibility of environmental migrant flows pressing on Europe also triggered a lot of interest, and all participants agreed that this might potentially evolve into a security concern for which the armed forces should be ready [99–102].

The officers' recommendations on how the impacts of CC on military assets should be addressed related to cutting-edge military technology and more informed personnel. No matter where they were from, every military officer suggested that the problem might be solved by modifying the current equipment to be more resilient to harsh weather and high temperatures. Additionally, they recommended that personnel should receive better training to adequately prepare them for the ever-changing operational environment and reduce the likelihood that they will be taken by surprise during a real incident [99–101].

When asked whether they believed that the appropriate means of response to environmental risks should be military or carried out by civilian authorities, military personnel emphasized that "environmental security issues are issues that should be resolved in cooperation with civil protection services but in any case, some of the issues that will arise due to environmental changes can only be resolved with the intervention of army" [102].

*4.5. Non-State Actors: Adelphi, SIPRI, EY, PRIO*

Participants in this category are affiliated with think tanks or organizations that support decision-making bodies and political leaders with their use of climate information. In response to the topic of who should benefit from security, it was implied that they should primarily be European citizens, but that focus should also be made on vulnerable populations residing in locations that may be affected by environmental phenomena [103,104]. One of them explained, "As a global leader in climate change concerns, Europe must persuade others to adopt more sustainable practices regarding carbon dioxide emissions as well as to alert them of potential threats in the future" [103]. The dominant opinion was that environmental events are insufficient on their own to be able to generate significant migrant movements, even if it was generally accepted that they can increase strain on already-stressed regions around the world by causing instability [105]; as the head of a major German think tank succinctly claimed, there are no "climate" or "environmental" migrants because "you are not leaving your country due of climatic pressures".

The primary issues for the nonstate actors' group include climate change along with several quick-onset environmental threats (earthquakes, tsunamis, etc.). However, the interviewees claimed that their perspectives and the scholarly ones do not agree on how these events relate to security issues. One of the respondents, a consultant and researcher for one of the largest climate security programs in Europe, highlighted that academic studies attempt to demonstrate the relationship between the environment and security through quantitative research, while his interactions with policymakers (as well as field studies) suggest the need to recognize the qualitative aspects of this connection [104]. Another respondent stressed that although the environment–security connection is frequently used by politicians to present something as an emergency issue, this practice usually is coupled with hidden security agendas [103].

The interviews revealed that the means we need to overcome the environmental issues that have security implications are funds and personnel, and not political will or security measures. The environmental aspects of the security concerns facing Europe must be handled by a diverse collection of practitioners, policymakers, researchers, and legislators [106]. A military response to environmental problems is needed only in a few particularly unique circumstances. The key challenge is to translate the political discourse on climate and environmental security into day-to-day activities; in other words, lower-level policymakers need to grasp these concepts since higher-level decision makers are already aware of them [105,106]. Therefore, to include environmental factors in their security missions, European ministries, agencies, and bodies require greater funding and people who are knowledgeable about environmental issues.

## 5. Discussion

In this study, we sought to identify the responses of the actors in the EU who influence the security agenda to the queries "security from what?", "security for whom?", and "security by which means?". We intend to address the EU's environmental security policies in this way. The responses covered the security actors' perceptions of environmental risks, their goals and responsibilities, and the tools and resources they have at their disposal to deal with environmental threats.

The responses provided by the respondents show how different each security actor's views are on what defines a security issue. In other words, to what extent could environmental issues—especially climate change—be considered a threat or security issue? Environmental issues vary in importance depending on the respondent: for some, they

are urgent and require immediate attention; for others, they are a threat multiplier and a source of additional pressure; and for others, they are irrelevant from a security standpoint. In addition, this study revealed how various initiatives and procedures are created based on how participants thought security should be utilized. As a result, some people only talk about national security, while others only talk about European security, some talk about the Transatlantic Alliance, and very few people regard the EU as a major player who will have an impact on the world. Finally, this study demonstrated how challenging it is to deal with environmental security concerns because of the competing missions and mandates of the actors and the lack of an environmental security issue guidance structure.

Our major findings about the environmental security components of the European security agenda—which will be substantiated in the following paragraphs—show that (a) despite the strong political will, the environmental security issues do not properly extend beyond the level of the European Commission or the European Parliament; (b) the opinions of midlevel and senior-level practitioners and policymakers are very different between organizations; (c) the various mandates and missions of each organization severely restrict the options and flexibility of organizations and agencies; and (d) more often than not, green projects in the security sector start with civilian personnel trying to understand political leadership's directives using whatever expertise they may have, and then the MoD's staff complying. Some of the conclusions reached from these findings are in line with prior research and are still valid, while others are contrary because the respondents' viewpoints may have changed, or the findings may have changed over time.

Concerning our first finding, despite the many political advances that have been achieved since Javier Solana attempted in 2008 to persuade Europe that addressing the climate issue is linked to maintaining global stability and ensuring European security [7], we found that there are inefficiencies in how political will is implemented into the the-day-to-day working of organizations that receive directives or instructions from the European Council, the European Parliament, and the European Commission. Such organizations are the EEAS, EDA, ESDC, EUISS, and MoDs of the member states. The European Commission, since 2008, has repeatedly included the environmental dimensions into the European security agenda in many different forms [8,9,13,69,70,72,107]. The "Climate Change and Defence Roadmap", launched in November 2020, served as an EEAS magnum opus, and our interviewees suggested that the Commission's general directorates now take environmental security considerations into account [84–88]. Nevertheless, challenges emerge as the relevant organizations seem unable to translate this dominant political commitment into systematic actions.

This is also corroborated by the fact that the European Parliament has little real influence on issues of environmental security. The environment–security nexus not only originates from the highest echelons of the Commission but also is raised within the ecosystem of the European Parliament [108]. Ten years ago, analysts claimed that the economic crisis had a considerable influence on the EU by limiting its internal environmental policy ambition, which in turn had a significant impact on the Union's ability to be an environmental leader and pioneer [109] (p. 320). Instead, current studies from the European Parliamentary Research Service claim that the European Green Deal, the idea of an "integrated approach for climate change and security", the roadmap for addressing climate change and defense, and programs such as the Strategic Compass are all aiming high for the future of the EU's security and climate action [110]. The participants in this study asserted that, along with hybrid and cyber threats, climate change must be seen as a new security concern, and that the European Parliament has introduced even more conflicting terminology, such as climate refugees [10,110–112]. Accordingly, this study contradicts the idea made by academics 2 years ago that the EU regards climate change as primarily posing a security risk to other, more vulnerable regions of the world, which might provide security issues for the EU [113]. The EU simultaneously affirms the political will required to address the hazards posed by climate change within its borders while also fully admitting its significance for countries in Africa and Asia.

Our second finding is that the opinions of midlevel and senior-level practitioners and policymakers are very different from organization to organization. As a result, the groups that influence the security agenda carry out their efforts considering specific environmental and climatic concerns. Middle- and high-ranking executives have very different perspectives on how to run their organizations to carry out political mandates in the ways they consider appropriate. Thus, our findings support previous research that has identified the emergence of a self-organized community of practice [14]. However, we also find that this community of practitioners is (still) uncoordinated and tends to develop projects based on their perceptions, with each agency emphasizing different environmental security issues/priorities. For instance, the European Union Satellite Centre appears to prioritize other issues above climate change, whereas the European Defense Agency is simultaneously developing the largest network for climate security in Europe and addressing climate change in three separate large-scale initiatives [114–116].

Our third finding is that each organization's different mandates and missions significantly limit the organizations' and agencies' alternatives and flexibility. Previous research has indicated that the mandates of organizations shape and, in some cases, limit the actions of organizations in matters of environmental security. Examples of such organizations include the EEAS, the Commission, and independent agencies such as the EDA, SatCen, ESDC, and EUISS. In 2014, Richard Youngs argued that "climate security has become one of the clearest examples of an issue that falls into the gaps between ministerial portfolios and institutional mandates" [12] (p. 49). In the same vein, Remling and Barnhoorn confirmed in 2021 that the institutional players in terms of climate security are made up of numerous EU bodies with various mandates [113]. In our study, the participants confirmed that in many cases, their mandates impose a lot of constraints when they outlined the activities of their institutions. They claimed that every political decision is a step toward implementing it for each organization in a way that is compatible with its mandate, which entails a lot of execution difficulties. To get an idea of these discrepancies, one should consider that climate *diplomacy* is a primary consideration at the EEAS, which is formally the European Union's foreign ministry; an environmental security *analyst* was hired at the EUISS, the continent's think tank; and a "climate security" *training module* was introduced at the ESDC, the EU's equivalent of a military university [80,81,93–96]. All these support those scholars arguing that the propensity of agencies is constrained by their mandates. However, it also demonstrates that we are unsure of whether these efforts reflect best practices. Therefore, the new element of our study is the necessity for either evaluation of these activities or coordination by a mechanism that would be able to see the big picture and promote an all-encompassing approach.

Our fourth finding is that "green" programs in the security sector begin with civilian employees attempting to comprehend political leadership's guidelines using whatever insight they may have, and then MoD officials following. Speaking with employees from various organizations about security concerns and how they view the risks and difficulties posed by environmental issues, particularly climate change, it becomes evident that any actions are carried out by an autonomous civilian community of practice. Due to their interaction in the broader Brussels area, diverse personnel who work for the European Commission, the EEAS, the EDA, or the ESDC used to know one another and help shape the European environmental security agenda. However, they are aware of the fact that no organization has planned their acts, and they lack formal scientific training [80–82,84,87,89–91,93,95,97]. Consequently, a significant endeavor is made to incorporate the environmental components into the security agenda without, however, conducting any evaluation of the projects' performance or adopting any measures to prevent any activity overlap. Numerous studies show that the armed forces are prepared and should launch green projects right away [117–123]. However, our participants claimed that the militaries of the member states do cooperate on other matters, but not on initiatives involving the environment or issues relating to climate change [89,90,99,102]. Additionally, after discussing with high-ranking military personnel from various MoDs, this study revealed that the security community takes part in the aforementioned community

of practice, but it plays a secondary role in determining the initiatives [99–102]. Without multilateral engagement between member states or a coordinating body in any Brussels institution, any efforts are undertaken independently at the national level [89,90]. That suggests the necessity to advance the community of practice into a network of professionals, both civilian and military personnel, managed by a formal body with demonstrated expertise in environmental security issues.

## 6. Conclusions

This study makes a theoretical and practical contribution to the field of environmental security. On a theoretical level, it became evident that it is useful to evaluate the problem in three axes to be able to perceive and assess whether an organization's security agenda appropriately encompasses the environmental dimensions of security challenges: who will benefit from the security, from what threats do those involved believe they should be protected, and what might be the means by which the actors involved would handle the issue. The methodology we employed in this study brought attention to the fact that the security agenda of an organization like the European Union is determined by numerous different actors, each of whom has a different perspective on all the aforementioned issues, which is why we have diverse approaches to problem solving. In our study, this methodology was applied for the first time, and it offers the chance for future researchers to use it to safely conclude related research questions.

Practically speaking, our research helped to highlight the difficulties that security actors in Europe confront and the fact that environmental issues are frequently handled insufficiently as a result of the variety of approaches that are used to address them. The political agenda appears strong, but it is insufficient since it is not coupled with enough recruitment and training and the creation of a structure that will oversee and guide the operations of the participating agencies. The projects require direction that is tailored to the agencies' mandates to ensure the most effective implementation of political directions while also coordinating to prevent duplications and omissions. This study also made clear the necessity for the security community to help shape the environmental security agenda rather than just react to events. The armed forces are demonstrating their ability and desire to adapt to address the national security issues that worry them while also addressing any environmental and climate challenges.

Access to the respondents and the time limits imposed by the nature of the research were the main constraints in this study. Due to the lack of an authority dealing with environmental issues in the security agencies we contacted, we had to look for the person who dealt with these issues. Thus, while many organizations described having undertaken specific environmental safety initiatives, it was difficult to identify the people behind these initiatives. This happened because not every organization had a specific department to perform this work, and usually, the people who had the environmental issues of the organization also had other responsibilities. Thus, the difficulty of access created the need to extend our research in time so that we could have a representative sample and be led to safe conclusions. Based on this fact, an important guideline for future researchers is, in any research, to search thoroughly and quickly for the workers who have taken on environmental safety issues in each safety organization they intend to investigate. Access is difficult, and often, the right people are hard to find.

**Author Contributions:** Conceptualization, D.K.; methodology, D.K.; validation, D.K. and I.B.; formal analysis, D.K. and I.B.; investigation, D.K.; resources, D.K.; data curation, D.K.; writing—original draft preparation, D.K.; writing—review and editing, I.B.; visualization, D.K.; supervision, I.B.; project administration, D.K.; funding acquisition, I.B. All authors have read and agreed to the published version of the manuscript.

**Funding:** This work was supported through a scholarship provided by the National Scholarship Foundation of Greece—IKY.

**Institutional Review Board Statement:** Not applicable.

**Informed Consent Statement:** Informed consent was obtained from all subjects involved in the study.

**Data Availability Statement:** Data are available on request due to ethical restrictions.

**Acknowledgments:** This paper is a section of a Ph.D. dissertation partly funded by the National Scholarship Foundation of Greece—IK.

**Conflicts of Interest:** There is no financial interest or benefit that has arisen from the direct applications of this research. The authors declare no conflict of interest.

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
