# Peer review of "Understanding the Environmental Security Perceptions of the European Union’s Security Actors"

_sustainability, doi:10.3390/su151713027_

Round 1

Reviewer 1 Report

The chosen topic is interesting and challenging. Unfortunately, this kind of study needs approval by an Institutional Review Board because the authors interacted with human subjects and changed data between institutions. An interview means interacting with people and assessing some attitudes and perceptions regarding understanding environmental security. Even if your university does not have such an entity, a Committee for Ethics might approve a study involving people from a few institutions. I understood that there is informed consent, but it is not enough from my point of view.

Reviewer 2 Report

I have the following recommendations:

1. Try to have a Literature review section; if you want, you can have subheadings related to historical background. Keep the traditional structure for scientific papers. I appreciate the references you included in the introduction and your second section. 

2. When you have references from 74-76, check the guidelines, you should put them between brackets like [74-76]. Check the entire paper

3. The methodology and the interviews are well detailed in section 3. Can you elaborate here on why the research duration was so high? More than a year. Also in Results can you detail the process of the interviews? Was it difficult? How was it conducted? What obstacles/barriers have you encountered? These should be highlighted

4. After Discussion, add a Conclusion section where you have to include the theoretical and practical implications of your research (usefulness, for whom, why, the novelty of your research), the limitations and future research directions. 

Reviewer 3 Report

Dear Authors,

Thank you for the opportunity to read the manuscript. 

There are a lot of interesting points but still some improvements are needed: 

1. Your theory finished with a common consensus that now the main discourse in EU is the climate security but not climate change, In this context why do you analyze environmental security if the discourse already changed. 

2. Can you provide the initial number of respondents that you planned to interview? How did you choose your respondents, what were the criteria? Can you make the lists with positions and codes? 

3. There are some grammar mistakes like line 446 Climate change is the top concern for all responders since it may cause problems.

4. Why you asked the respondent from Prio if it is not EU territory?

5. The section on discussion needs some shift to conclusions with policy implications and limitations. 

All the best

Round 2

Reviewer 1 Report

As I said, the study is interesting but has some ethical issues.

Please, provide us with the number and date when your home University's Research Ethics and Integrity Committee permitted this study. This committee is the equivalent of IRB in the EU Universities. Also, please tell us if there is a signed Data transfer and use agreement between your home university and the institutions from where the participants in your research are.

Even the participants from your study wanted to answer, they represent some institutions. Their answers are linked to an institution. Moreover, you can identify their responses if the interview was in person. 

Author Response

Dear Reviewer   Thank you for your comment. The research's protocol -and the questionnaire used in- this research was submitted to the University of the Aegean’s Research Ethics and Integrity Committee (E.H.D.E.) in January 2021. Under Greek Legislation (Law 4521/2018, Chapter E’), E.H.D.E. is obliged to assess and give formal approval for research involving human subjects only in those cases that the research is funded by national or EU funds (i.e. it is part of a funded project administered by the university). For non-funded research, as it is our case, it rests at the researchers’ discretion to contact the Research Ethics’ committee E.H.D.E. and ask for an assessment. In this case, as stipulated by the Greek law, the Research Ethics’ committee does NOT “approve or disapprove” the research format but simply ‘opines’ on its suitability. In this latter case, the Research Ethics’ committee does NOT issue an official decision but simply informs the researchers about its views on the ethical appropriateness of the research proposal. This has been our case, where the Committee informed us, in January 2021, that it ‘opines’ than our research protocol/questionnaire do not runs counter to the University of the Aegean’s Research Ethics & Integrity Regulations.   Thus, in the case of your journal’s “Institutional Review Board Statement” declaration, the appropriate entry should read: ‘No official approval required under Greek Law 4521/2018”.   I hope this clarify the issue  

Kind regards  

Dr. Iosif Botetzagias

Reviewer 3 Report

Dear Authors,

Thank you for the updated article. It can be published in a Sustainability journal.

All the best 

Author Response

Thank you very much for your feedback!

Respectfully,

The authors

Round 3

Reviewer 1 Report

Dear authors,

Thank you for indicating the number and day you asked the Ethics and Integrity Committee to approve this study. This information shows me that the approval was before starting your investigation. For this reason, I changed my decision. I just wanted to know that your Committee was informed about your study. 

As I said from the beginning, the topic of your paper is excellent, and I appreciate your courage to approach such a topic. Those people that answer your questionnaire/interview represent institutions; from this point of view, there are some risks, and I will keep my opinion.

Even though the general wants to see his name on the paper, he did not sign any document to assure you that he will keep his option. Please consider this thing next time when you approach such a sensitive topic.

Thank you.

Author Response

Dear reviewer,

thank you very much for your comments and your directions for this paper. We appreciate your effort and we understand that many responses of the interviewees represent institutions which contain several risks. 

We have take specific measures to conduct this research according to the Greek law and with the consent of all the people who are engaged in this research. All communications with the interviewees involved are at our disposal in case there is any issue from any organization.

Again, thank you for your comments and feedback.

Respectfully,

The authors